# ATTACKING THE MADRY DEFENSE MODEL WITH $L_1$-BASED ADVERSARIAL EXAMPLES

**Yash Sharma[1] and Pin-Yu Chen[2]**

[1]The Cooper Union, New York, NY 10003, USA
[2]IBM Research, Yorktown Heights, NY 10598, USA
sharma2@cooper.edu, pin-yu.chen@ibm.com

## ABSTRACT

The Madry Lab recently hosted a competition designed to test the robustness of their adversarially trained MNIST model. Attacks were constrained to perturb each pixel of the input image by a scaled maximal $L_\infty$ distortion $\epsilon = 0.3$. This decision discourages the use of attacks which are not optimized on the $L_\infty$ distortion metric. Our experimental results demonstrate that by relaxing the $L_\infty$ constraint of the competition, the **e**lastic-net **a**ttack to **d**eep neural networks (EAD) can generate transferable adversarial examples which, despite their high average $L_\infty$ distortion, have minimal visual distortion. These results call into question the use of $L_\infty$ as a sole measure for visual distortion, and further demonstrate the power of EAD at generating robust adversarial examples.

## 1 INTRODUCTION

Deep neural networks (DNNs) achieve state-of-the-art performance in various tasks in machine learning and artificial intelligence, such as image classification, speech recognition, machine translation and game-playing. Despite their effectiveness, recent studies have illustrated the vulnerability of DNNs to adversarial examples Szegedy et al. (2013); Goodfellow et al. (2015). For instance, a carefully designed perturbation to an image can lead a well-trained DNN to misclassify. Even worse, effective adversarial examples can also be made virtually indistinguishable to human perception. Adversarial examples crafted to evade a specific model can even be used to mislead other models trained for the same task, exhibiting a property known as transferability.

To address this problem, the adversarial robustness of neural networks was studied through the lens of robust optimization in Madry et al. (2017), leading to an adversarially robust model for image classification we term as the "Madry Defense Model". An attack challenge[1] was proposed for the MNIST dataset to test the defense, however, attacks were constrained to perturb each pixel by at most $\epsilon = 0.3$, a scaled maximal $L_\infty$ distortion. This rule greatly reduces the power of attacks which are not optimized for the $L_\infty$ distortion metric, and imposes an unrealistic constraint on the attacker.

To justify the limitation of the $L_\infty$ constraint, we conduct extensive experiments on the Madry Defense Model to investigate the transferability properties of the state-of-the-art adversarial attacks without the $L_\infty$ constraint. We find that $L_1$-based adversarial examples generated by EAD, which is short for **E**lastic-net **A**ttack to **D**NNs Chen et al. (2017), readily transfer in both the targeted and non-targeted cases, and despite the high $L_\infty$ distortion, the visual distortion on the adversarial examples is minimal. These results call into question the use of $L_\infty$ to quantify visual distortion, and further demonstrate the state-of-the-art transferability properties of EAD.

## 2 EXPERIMENT SETUP

The Madry Defense Model is a sufficiently high capacity network trained against the strongest possible adversary, which they deem to be Projected Gradient Descent (PGD) starting from random

---

[1]https://github.com/MadryLab/mnist_challenge

Table 1: Comparison of tuned PGD, I-FGM, C&W, and EAD adversarial examples at various confidence levels. ASR means attack success rate (%). The distortion metrics are averaged over successful examples.

| Attack Method | Confidence | Targeted | | | | Non-Targeted | | | |
|---|---|---|---|---|---|---|---|---|---|
| | | ASR | $L_1$ | $L_2$ | $L_\infty$ | ASR | $L_1$ | $L_2$ | $L_\infty$ |
| PGD | None | 68.5 | 188.3 | 8.947 | 0.6 | 99.9 | 270.5 | 13.27 | 0.8 |
| I-FGM | None | 75.1 | 144.5 | 7.406 | 0.915 | 99.8 | 199.4 | 10.66 | 0.9 |
| C&W | 10 | 1.1 | 34.15 | 2.482 | 0.548 | 4.9 | 23.23 | 1.702 | 0.424 |
| | 30 | 69.4 | 68.14 | 4.864 | 0.871 | 71.3 | 51.04 | 3.698 | 0.756 |
| | 50 | 92.9 | 117.45 | 8.041 | 0.987 | 99.1 | 78.65 | 5.598 | 0.937 |
| | 70 | 34.8 | 169.7 | 10.88 | 0.994 | 99 | 119.4 | 8.097 | 0.99 |
| EAD | 10 | 27.4 | 25.79 | 3.209 | 0.876 | 39.9 | 19.19 | 2.636 | 0.8 |
| | 30 | 85.8 | 49.64 | 5.179 | 0.995 | 94.5 | 34.28 | 4.192 | 0.971 |
| | 50 | 98.5 | 93.46 | 7.711 | 1 | 99.6 | 57.68 | 5.839 | 0.999 |
| | 70 | 67.2 | 148.9 | 10.36 | 1 | 99.8 | 90.84 | 7.719 | 1 |

perturbations around the natural examples Madry et al. (2017). For the MNIST model, 40 iterations of PGD were run, with a step size of 0.01. Gradient steps were taken in the $L_\infty$ norm. The network was trained and evaluated against perturbations of size no greater than $\epsilon = 0.3$. We experimentally show in the supplementary material that this is the maximum $\epsilon$ for which the Madry Defense Model can be successfully adversarially trained.

We test the transferability properties of attacks in both the targeted case and non-targeted case against the adversarially trained model. We test the ability for attacks to generate transferable adversarial examples using a single undefended model as well as an ensemble of undefended models, where the ensemble size is set to 3. We expect that if an adversarial example remains adversarial to multiple models, it is more likely to transfer to other unseen models Liu et al. (2016); Papernot et al. (2016). We discuss results transferring from an ensemble in the following sections, and provide results transferring from a single undefended model in the supplementary material. The undefended models we use are naturally trained networks of the same architecture as the defended model; the architecture was provided in the competition. For generating adversarial examples, we compare the optimization-based approach and the fast gradient-based approach using EAD and PGD, respectively.

EAD generalizes the state-of-the-art C&W $L_2$ attack Carlini & Wagner (2017) by performing elastic-net regularization, linearly combining the $L_1$ and $L_2$ penalty functions Chen et al. (2017). The hyperparameter $\beta$ controls the trade-off between $L_1$ and $L_2$ minimization. We test EAD in both the general case and the special case where $\beta$ is set to 0, which is equivalent to the C&W $L_2$ attack. We tune the $\kappa$ hyperparameter, which controls the necessary margin between the predicted probability of the target class and that of the rest, in our experiments. Full detail on the implementation is provided in the supplementary material.

For $L_\infty$ attacks, which we will consider, fast gradient methods (FGM) use the sign of the gradient $\nabla J$ of the training loss $J$ with respect to the input for crafting adversarial examples Goodfellow et al. (2015). Iterative fast gradient methods (I-FGM) iteratively use FGM with a finer distortion, followed by an $\epsilon$-ball clipping Kurakin et al. (2016). We note that this is essentially projected gradient descent on the negative loss function. We test PGD both with and without random starts, and as the classic I-FGM formulation does not include random starts, PGD without random starts shall be termed I-FGM for the remainder of the paper. We perform grid search to tune $\epsilon$, which controls the allowable $L_\infty$ distortion. Full detail on the implementation is provided in the supplementary material.

## 3 EXPERIMENT RESULTS

In our experiment, 1000 random samples from the MNIST test set were used. For the targeted case, a target class that is different from the original one was randomly selected for each input image. The results of the $\beta$ tuning for EAD is provided in the supplementary material. It is observed that the highest attack success rate (ASR) in both the targeted and non-targeted cases was yielded at $\beta = 0.01$. This was in fact the largest $\beta$ tested, indicating the importance of minimizing the $L_1$ distortion for generating transferable adversarial examples. Furthermore, as can be seen in the supplementary

material, the improvement in ASR with increasing $\beta$ is more significant at lower $\kappa$, indicating the importance of minimizing the $L_1$ distortion for generating transferable adversarial examples with minimal visual distortion. For PGD and I-FGM, $\epsilon$ was tuned from 0.1 to 1.0 at 0.1 increments, full results of which are provided in the supplementary material.

In Table 1, the results for tuning $\kappa$ for C&W and EAD at $\beta = 0.01$ are provided, and are presented with the results for PGD and I-FGM at the lowest $\epsilon$ values at which the highest ASR was yielded, for comparison. It is observed that in both the targeted case and the non-targeted case, EAD outperforms C&W at all $\kappa$. Furthermore, in the targeted case, at the optimal $\kappa = 50$, EAD's adversarial examples surprisingly have lower average $L_2$ distortion.

In the targeted case, EAD outperforms PGD and I-FGM at $\kappa = 30$ with much lower $L_1$ and $L_2$ distortion. In the non-targeted case, PGD and I-FGM yield similar ASR at lower $L_\infty$ distortion. However, we argue that in the latter case the drastic increase in induced $L_1$ and $L_2$ distortion of PGD and I-FGM to generate said adversarial examples indicates greater visual distortion, and thus that the generated examples are less adversarial in nature. We examine this claim in the following section.

## 4 VISUAL COMPARISON

Adversarial examples generated by EAD and PGD were analyzed in the non-targeted case, to understand if the successful examples are visually adversarial. A similar analysis in the targeted case is provided in the supplementary material. We find that adversarial examples generated by PGD even at low $\epsilon$ such as 0.3, at which the attack performance is weak, have visually apparent noise.

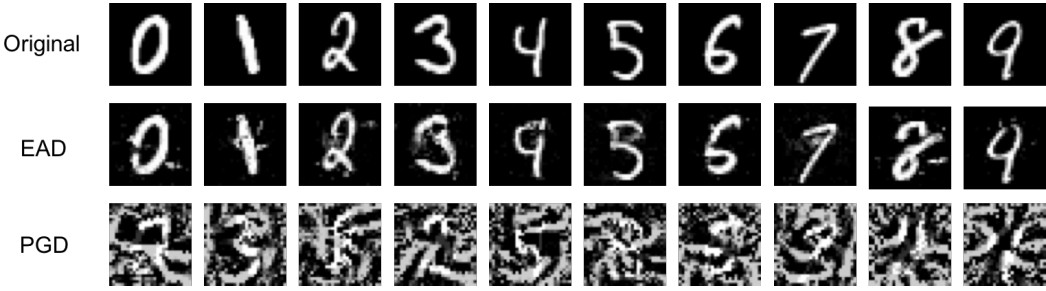

Figure 1: Visual illustration of adversarial examples crafted in the non-targeted case by EAD and PGD with similar average $L_\infty$ distortion.
.

In Figure 1, adversarial examples generated by EAD are directly compared to those generated by PGD with similar average $L_\infty$ distortion. EAD tuned to the optimal $\beta$ (0.01) was used to generate adversarial examples with $\kappa = 10$. As can be seen in Table 1, the average $L_\infty$ distortion under this configuration is 0.8. Therefore, adversarial examples were generated by PGD with $\epsilon = 0.8$. Clearly, performing elastic-net minimization aids in minimizing visual distortion, even when the $L_\infty$ distortion is large. Furthermore, these results also emphasize the issues in solely using $L_\infty$ to measure visual distortion, and how setting such a distortion constraint for adversarial training is not sufficient for characterizing the subspaces of visually similar adversarial examples.

## 5 CONCLUSION

The Madry Lab developed a defense model by focusing on training a sufficiently high-capacity network and using the strongest possible adversary. By their estimations, this adversary was PGD, which they motivate as the strongest attack utilizing the local first-order information about the network. Through extensive experiments, we demonstrate that EAD is able to outperform PGD in transferring in the targeted case. We also show that EAD is able to generate much less visually distorted transferable adversarial examples than PGD with comparable $L_\infty$ distortion, due to the drastic reduction in $L_1$ and $L_2$ distortion. These results demonstrate the power of EAD, particularly in its transferability capabilities. Furthermore, these results indicate the drawbacks of using $L_\infty$ as the sole distortion metric, and suggest incorporating $L_1$ or $L_2$ to complement the subspace analysis of adversarial examples for possible defenses such as adversarial training and adversary detection.

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

## 6 SUPPLEMENTARY MATERIAL

### 6.1 RETRAINING THE MADRY DEFENSE MODEL (TABLE 2)

The MNIST model released by the Madry Lab is adversarially trained using PGD with $\epsilon = 0.3$. Therefore, it is expected that the existing Madry Defense Model performs poorly against PGD attacks with $\epsilon > 0.3$.

In order to rectify this issue, the Madry Defense Model was trained with and evaluated against a PGD adversary with $\epsilon$ tuned from 0.1 to 1.0 under 0.1 increments. As suggested in Madry et al. (2017), the PGD attack was run for 40 iterations, and to account for varying $\epsilon$, the stepsize was set to $2\epsilon/40$.

The adversarial retraining results are shown in Table 2. These results suggest that the Madry Defense Model can not be successfully adversarially trained using a PGD adversary with $\epsilon > 0.3$. This is understandable as with such large $\epsilon$, the visual distortion is clearly perceptible.

### 6.2 ATTACK DETAILS

The targeted attack formulations are discussed below, non-targeted attacks can be implemented in a similar fashion. We denote by $\mathbf{x}_0$ and $\mathbf{x}$ the original and adversarial examples, respectively, and denote by $t$ the target class to attack.

#### 6.2.1 EAD

EAD generalizes the state-of-the-art C&W $L_2$ attack Carlini & Wagner (2017) by performing elastic-net regularization, linearly combining the $L_1$ and $L_2$ penalty functions Chen et al. (2017). The formulation is as follows:

$$\text{minimize}_\mathbf{x} \ c \cdot f(\mathbf{x}, t) + \beta \|\mathbf{x} - \mathbf{x}_0\|_1 + \|\mathbf{x} - \mathbf{x}_0\|_2^2$$
$$\text{subject to} \ \ \mathbf{x} \in [0, 1]^p, \tag{1}$$

where $f(x, t)$ is defined as:

$$f(\mathbf{x}, t) = \max\{\max_{j \neq t}[\mathbf{Logit}(\mathbf{x})]_j - [\mathbf{Logit}(\mathbf{x})]_t, -\kappa\}, \tag{2}$$

By increasing $\beta$, one trades off $L_2$ minimization for $L_1$ minimization. When $\beta$ is set to 0, EAD is equivalent to the C&W $L_2$ attack. By increasing $\kappa$, one increases the necessary margin between the predicted probability of the target class and that of the rest. Therefore, increasing $\kappa$ improves transferability but compromises visual quality.

We implement 9 binary search steps on the regularization parameter $c$ (starting from 0.001) and run $I = 1000$ iterations for each step with the initial learning rate $\alpha_0 = 0.01$. For finding successful adversarial examples, we use the ADAM optimizer for the C&W attack and implement the projected FISTA algorithm with the square-root decaying learning rate for EAD Kingma & Ba (2014); Beck & Teboulle (2009).

#### 6.2.2 PGD

Fast gradient methods (FGM) use the gradient $\nabla J$ of the training loss $J$ with respect to $\mathbf{x}_0$ for crafting adversarial examples Goodfellow et al. (2015). For $L_\infty$ attacks, which we will consider, $\mathbf{x}$ is crafted by

$$\mathbf{x} = \mathbf{x}_0 - \epsilon \cdot \text{sign}(\nabla J(\mathbf{x}_0, t)), \tag{3}$$

where $\epsilon$ specifies the $L_\infty$ distortion between $\mathbf{x}$ and $\mathbf{x}_0$, and $\text{sign}(\nabla J)$ takes the sign of the gradient.

Iterative fast gradient methods (I-FGM) were proposed in Kurakin et al. (2016), which iteratively use FGM with a finer distortion, followed by an $\epsilon$-ball clipping. We note that this is essentially projected gradient descent on the negative loss function.

Consistent with the implementation in Madry et al. (2017), 40 steps were used. For PGD without random starts, or I-FGM, the step size was set to be $\epsilon/40$, which has been shown to be an effective attack setting in Tramèr et al. (2017). For PGD with random starts, the step size was set to be $2\epsilon/40$ in order to allow all pixels to reach all possible values.

### 6.3 PGD Adversarial Examples (Figure 2)

In Figure 2, a selected instance (a hand-written '1') of adversarial examples crafted by PGD at $\epsilon$ tuned from 0.1 to 1.0 at 0.1 increments is shown. As PGD simply operates on a $L_\infty$ distortion budget, as $\epsilon$ increases, the amount of noise induced to all of the pixels in the image increases. This noise is visually obvious even at low $\epsilon$ such as 0.3, where the adversarial example does not even successfully transfer. At higher $\epsilon$ values, PGD appears to change the '1' to a '3'. However, under such large visual distortion, it is not entirely clear.

### 6.4 Visual Comparison in the Targeted Case (Figure 3)

In Figure 3, adversarial examples generated by EAD are directly compared to those generated by PGD with similar average $L_\infty$ distortion in the targeted case. Adversarial examples generated by EAD tuned to the optimal $\beta$ (0.01) at $\kappa = 10$ have an average $L_\infty$ distortion of 0.876. Therefore, these examples were compared to those generated by PGD with $\epsilon = 0.8$ and $\epsilon = 0.9$. It can be seen that further distortion is needed to generate transferable adversarial examples in the targeted case, however, the benefit that elastic-net minimization applies for improving visual quality is still clearly evident.

### 6.5 Single Model Results (Table 3)

In Table 3, the results of tuning $\kappa$ for C&W and EAD at the optimal $\beta$ (0.01) are provided, and are presented with the results for PGD and I-FGM at the lowest $\epsilon$ where the highest ASR was yielded, for comparison. Alike the results obtained when using an ensemble of models, it is observed that EAD outperforms C&W at all $\kappa$. In the targeted case, at $\kappa = 50$, both EAD and C&W outperform PGD and I-FGM with much lower $L_1$ and $L_2$ distortion. Furthermore, at $\kappa \geq 50$, EAD's adversarial examples unexpectedly have lower average $L_2$ distortion than C&W's. In the non-targeted case, PGD and I-FGM outperform EAD in terms of ASR. However, as can be inferred by the large $L_1$ and $L_2$ distortion, the perturbations are visually perceptible.

### 6.6 Beta Search for EAD (Tables 4 and 5)

In Table 4, the results of tuning $\beta$ when generating adversarial examples with EAD attacking an ensemble of models are presented. In Table 5, the results of tuning $\beta$ when generating adversarial examples with EAD attacking a single model are presented. In both cases, one can see that at the optimal $\kappa$ in terms of ASR, tuning $\beta$ plays relatively little effect in improving the ASR. However, at lower $\kappa$, which is applicable when visual quality is important, increasing $\beta$ plays a rather significant role in improving the ASR. This suggests that for generating transferable adversarial examples with minimal visual distortion, minimizing the $L_1$ distortion is more important than minimizing $L_2$, or, by extension, $L_\infty$.

### 6.7 Epsilon Search for PGD and I-FGM (Figures 4 and 5)

In Figure 4, the results of tuning $\epsilon$ when generating adversarial examples with PGD and I-FGM attacking an ensemble of models are shown. In Figure 5, the results of tuning $\epsilon$ when generating adversarial examples with PGD and I-FGM attacking a single model are shown. The results simply show that by adding large amounts of visual distortion to the original image, an attacker can cause a target defense model to misclassify. However, when compared to EAD, the adversarial examples of PGD and I-FGM are less transferable and, when successful, the perturbations are more visually perceptible, as indicated by the drastic increase in the $L_1$ and $L_2$ distortion at high $\epsilon$.

Table 2: Results of training the Madry Defense Model with a PGD adversary constrained under varying $\epsilon$. 'Nat Test Accuracy' denotes accuracy on classifying original images. 'Adv Test Accuracy' denotes accuracy on classifying adversarial images generated by the same PGD adversary as was used for training.

| Epsilon | Nat Test Accuracy | Adv Test Accuracy |
|---------|-------------------|-------------------|
| 0.1 | 99.38 | 95.53 |
| 0.2 | 99 | 92.65 |
| 0.3 | 98.16 | 91.14 |
| 0.4 | 11.35 | 11.35 |
| 0.5 | 11.35 | 11.35 |
| 0.6 | 11.35 | 11.35 |
| 0.7 | 10.28 | 10.28 |
| 0.8 | 11.35 | 11.35 |
| 0.9 | 11.35 | 11.35 |
| 1 | 11.35 | 11.35 |

ε = 0.1  ε = 0.2  ε = 0.3  ε = 0.4  ε = 0.5  ε = 0.6  ε = 0.7  ε = 0.8  ε = 0.9 ε = 1.0

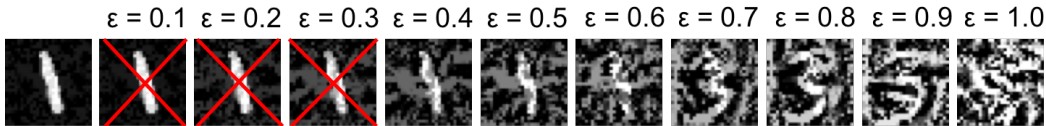

Figure 2: Visual illustration of adversarial examples crafted by PGD with $\epsilon$ tuned from 0.1 to 1.0 at 0.1 increments. The leftmost image is the original example. With $\epsilon \leq 0.3$, the adversarial examples were unsuccessful at transferring to the Madry Defense Model.
.

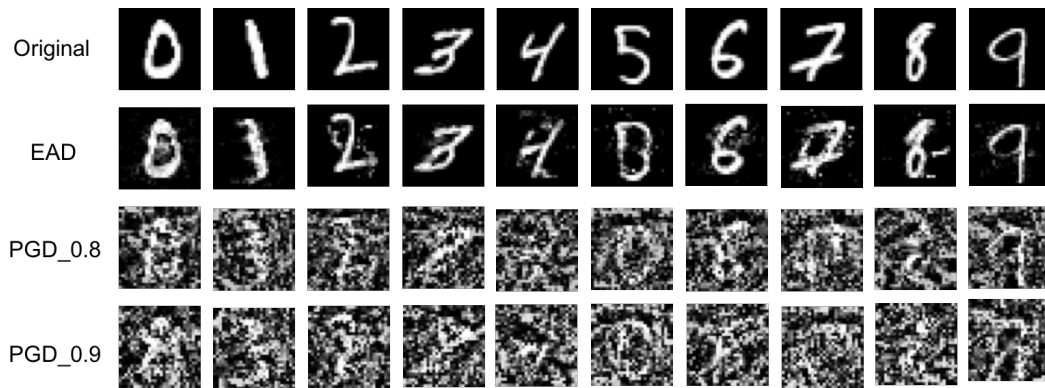

Figure 3: Visual illustration of adversarial examples crafted in the targeted case by EAD and PGD with similar average $L_\infty$ distortion. As the adversarial examples generated by EAD have an average $L_\infty$ distortion of 0.876, adversarial examples generated by PGD with $\epsilon = 0.8$ and $\epsilon = 0.9$ are shown.
.

Table 3: Comparison of tuned PGD, I-FGM, C&W, and EAD adversarial examples at various confidence levels generated from attacking a single model. ASR means attack success rate (%). The distortion metrics are averaged over successful examples.

| Attack Method | Confidence | Targeted | | | | Non-Targeted | | | |
|---|---|---|---|---|---|---|---|---|---|
| | | ASR | $L_1$ | $L_2$ | $L_\infty$ | ASR | $L_1$ | $L_2$ | $L_\infty$ |
| PGD | None | 41.2 | 214.5 | 10.18 | 0.7 | 99.7 | 237.1 | 11.52 | 0.7 |
| I-FGM | None | 39.2 | 133.6 | 6.801 | 0.87 | 99.3 | 160.8 | 8.395 | 0.7 |
| C&W | 10 | 0.1 | 17.4 | 1.112 | 0.19 | 1.8 | 12.57 | 0.801 | 0.156 |
| | 30 | 4.8 | 47.9 | 3.235 | 0.69 | 8.5 | 35.5 | 2.371 | 0.53 |
| | 50 | 51.5 | 81.67 | 5.455 | 0.913 | 46.9 | 59.18 | 3.873 | 0.765 |
| | 70 | 75.8 | 114.6 | 7.499 | 0.986 | 89.7 | 82.85 | 5.375 | 0.925 |
| EAD | 10 | 3.6 | 13.21 | 2.22 | 0.848 | 11.8 | 10.91 | 1.856 | 0.725 |
| | 30 | 32.7 | 33.75 | 3.735 | 0.963 | 44.9 | 25.98 | 3.129 | 0.906 |
| | 50 | 67.2 | 59.05 | 5.404 | 0.998 | 81 | 41.89 | 4.319 | 0.978 |
| | 70 | 82.7 | 91.15 | 7.227 | 1 | 94.7 | 61.69 | 5.572 | 0.997 |

Table 4: Analysis of adversarial examples generated by EAD tuned with various beta values and confidence levels when attacking an ensemble of models. ASR means attack success rate (%). The distortion metrics are averaged over successful examples.

| Beta | Confidence | Targeted | | | | Non-Targeted | | | |
|---|---|---|---|---|---|---|---|---|---|
| | | ASR | $L_1$ | $L_2$ | $L_\infty$ | ASR | $L_1$ | $L_2$ | $L_\infty$ |
| 1e-4 | 10 | 1.4 | 31.67 | 2.435 | 0.614 | 5.7 | 23.52 | 1.795 | 0.487 |
| | 30 | 67.5 | 64.84 | 4.526 | 0.894 | 71 | 50.58 | 3.641 | 0.812 |
| | 50 | 98.7 | 107.3 | 7.094 | 0.997 | 98.9 | 76.68 | 5.361 | 0.965 |
| | 70 | 74.3 | 163.6 | 9.993 | 1 | 99.8 | 112.9 | 7.466 | 0.998 |
| 1e-3 | 10 | 4.8 | 26.31 | 2.489 | 0.701 | 14.1 | 20.71 | 2.087 | 0.622 |
| | 30 | 75.5 | 59.62 | 4.642 | 0.941 | 83.5 | 43.16 | 3.69 | 0.871 |
| | 50 | 98.6 | 99.98 | 7.185 | 0.999 | 99.3 | 69.56 | 5.411 | 0.979 |
| | 70 | 73 | 155.2 | 10 | 1 | 99.8 | 106.4 | 7.526 | 0.999 |
| 1e-2 | 10 | 27.4 | 25.79 | 3.209 | 0.876 | 39.9 | 19.19 | 2.636 | 0.8 |
| | 30 | 85.8 | 49.64 | 5.179 | 0.995 | 94.5 | 34.28 | 4.192 | 0.971 |
| | 50 | 98.5 | 93.46 | 7.711 | 1 | 99.6 | 57.68 | 5.839 | 0.999 |
| | 70 | 67.2 | 148.9 | 10.36 | 1 | 99.8 | 90.84 | 7.719 | 1 |

Table 5: Analysis of adversarial examples generated by EAD tuned with various beta values and confidence levels when attacking a single model. ASR means attack success rate (%). The distortion metrics are averaged over successful examples.

| Beta | Confidence | Targeted | | | | Non-Targeted | | | |
|---|---|---|---|---|---|---|---|---|---|
| | | ASR | $L_1$ | $L_2$ | $L_\infty$ | ASR | $L_1$ | $L_2$ | $L_\infty$ |
| 1e-4 | 10 | 0.1 | 16.03 | 1.102 | 0.186 | 2 | 12.68 | 0.867 | 0.193 |
| | 30 | 2.8 | 43.02 | 2.88 | 0.658 | 9.8 | 34.21 | 2.33 | 0.558 |
| | 50 | 36.8 | 75.5 | 4.76 | 0.891 | 41.8 | 58.06 | 3.723 | 0.773 |
| | 70 | 75.1 | 107.7 | 6.629 | 0.988 | 84.6 | 81.18 | 5.063 | 0.924 |
| 1e-3 | 10 | 0.3 | 16.23 | 1.45 | 0.379 | 2.5 | 11.79 | 1.037 | 0.286 |
| | 30 | 7.5 | 39.42 | 3.025 | 0.739 | 15.4 | 30.13 | 2.453 | 0.658 |
| | 50 | 49.8 | 69.66 | 4.892 | 0.942 | 53.3 | 52.67 | 3.779 | 0.839 |
| | 70 | 78 | 99.68 | 6.711 | 0.996 | 88 | 74.47 | 5.096 | 0.951 |
| 1e-2 | 10 | 3.6 | 13.21 | 2.22 | 0.848 | 11.8 | 10.91 | 1.856 | 0.725 |
| | 30 | 32.7 | 33.75 | 3.735 | 0.963 | 44.9 | 25.98 | 3.129 | 0.906 |
| | 50 | 67.2 | 59.05 | 5.404 | 0.998 | 81 | 41.89 | 4.319 | 0.978 |
| | 70 | 82.7 | 91.15 | 7.227 | 1 | 94.7 | 61.69 | 5.572 | 0.997 |

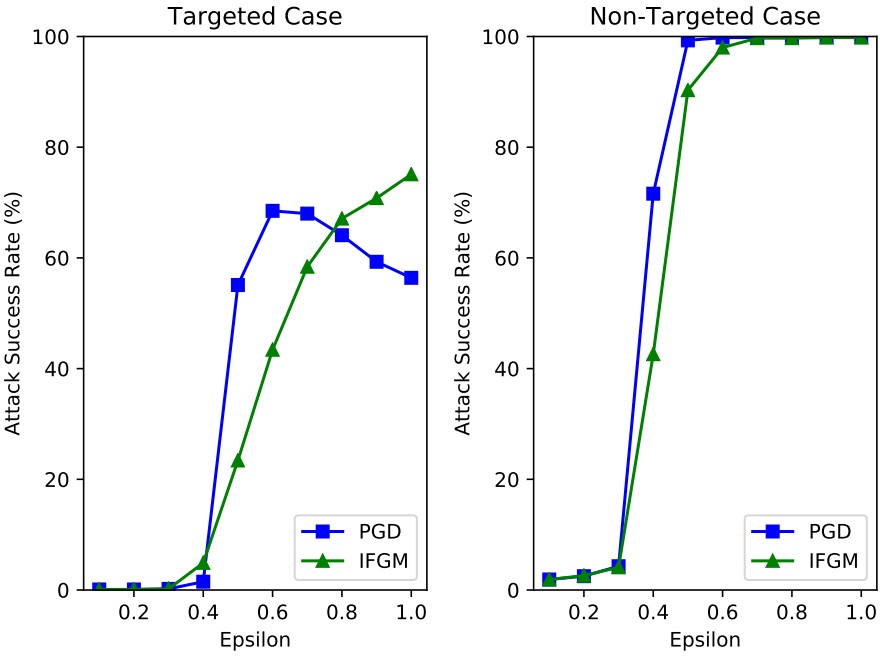

Figure 4: Attack transferability of adversarial examples generated by PGD and I-FGM with various $\epsilon$ values when attacking an ensemble of models.

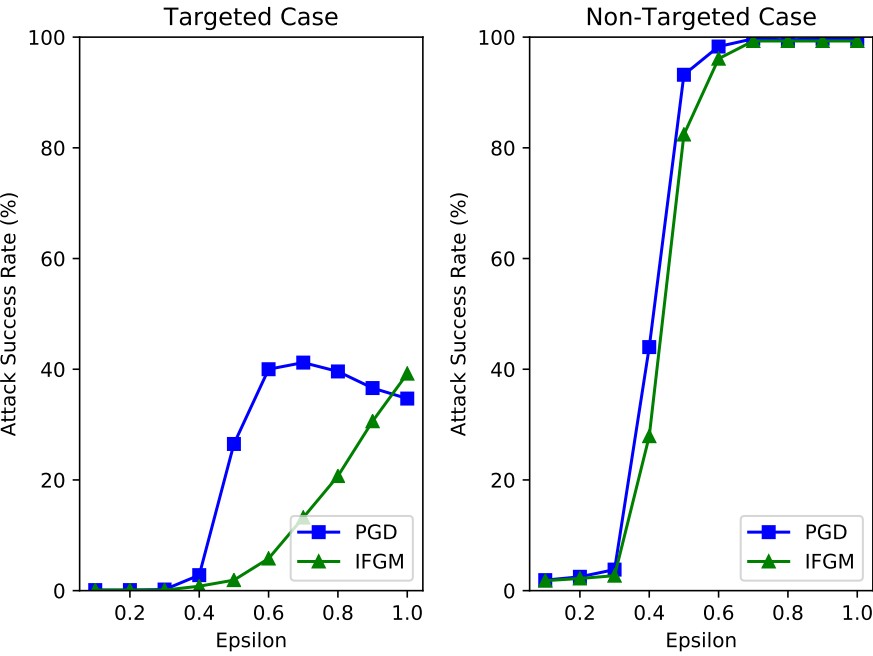

Figure 5: Attack transferability of adversarial examples generated by PGD and I-FGM with various $\epsilon$ values when attacking a single model.

