# OpenReview forum: "Attacking the Madry Defense Model with $L_1$-based Adversarial Examples"
_ICLR.cc/2018/Workshop — Accept_

### Official Review · AnonReviewer2 · 2018-03-07
**interesting experiments on the adversarial attacks**

**Rating:** 6
**Confidence:** 3

**Review:**

This paper emphazises the issue of using L_\infty norm when generating adversarial examples: the obtained attacks are less efficient that those using L_1.
This paper also provides experiments suggesting that adversarial examples generating by PGD algorithm using L_\infty norm are less visualy similar than those generated by EAD algorithm.

I suggest to authors to provide experiments to compare the visual similarity of adversarial examples generated using L_\infty and L_1 norm. For increasing the clarity of the paper, I also suggest to explain what is a non-targeted attack.

---

### Official Review · AnonReviewer1 · 2018-03-09
**Questioning L_\infty-bounded adversarial perturbations**

**Rating:** 7
**Confidence:** 5

**Review:**

The authors consider L_1 and L_2-bounded adversarial perturbations generated by Elastic-net Attack to DNNs (EAD) and observe that EAD gives better visually imperceptible and transferable attacks, despite high L_\infty distortion. Thus, they argue that adversarial training against only L_\infty-bounded adversarial attacks such as PGD (Projected Gradient Descent) and FGM (Fast Gradient Method) may not be sufficient.

In the targeted case, EAD achieves the same or better attack success rate than PGD and iterative-FGM with much lower L_1 and L_2 distortion. In the non-targeted case, PGD and iterative-FGM give similar attack success rate at lower L_\infty distortion. However, in this case, the authors observe in Figure 1 that the examples generated by EAD have less visual distortion or noise.

Overall, this paper has two contributions. First, it questions the choice of L_\infty-bounded attacks to model visual imperceptible adversarial perturbations. Second, it compares the visual distortion of the L_1/L_2-bounded attacks generated by EAD and the L_\infty-bounded attacks generated by PGD and iterative-FGM. I think both are decent contributions, worth a place in ICLR workshop.

---

### Official Review · AnonReviewer3 · 2018-03-12
**An interesting read, few details missing**

**Rating:** 7
**Confidence:** 1

**Review:**

The paper discusses a different attack strategy to assess the adversarial attack resistant model as proposed in Madry et al. Madry et al. suggested generating the adversarial samples by perturbing each pixel with a maximum L_\infty distortion of 0.3. The authors uses elastic net distortion indeed and show that this can produce transferable adversarial samples with higher L_\infty distortion yet visually similar to the original images. The proposed approach  is a generalization of the L_2 attack discussed by Carlini and Wagner. The paper is nicely written and presents the concepts and results well.

Minor comments:
Define Confidence in Table 1
Please explain t in Eq. 1 and 2.

---

### Decision · Program_Chairs · 2018-03-20
**ICLR 2018 Workshop Acceptance Decision**

**Decision:**

Accept

**Comment:**

Congratulations, your paper was accepted to the ICLR workshop.